# COVID-19 Booster Doses: A Multi-Center Study Reflecting Healthcare Providers’ Perceptions

**DOI:** 10.3390/vaccines11061061

**Published:** 2023-06-04

**Authors:** Hager Salah, Israa Sinan, Omar Alsamani, Lamyaa Samir Abdelghani, May Hassan ElLithy, Nazar Bukamal, Huda Jawad, Raghda R. S. Hussein, Marwa O. Elgendy, Al shaimaa Ibrahim Rabie, Doaa Mahmoud Khalil, Amira S. A. Said, Mohammad M. AlAhmad, Azza Khodary

**Affiliations:** 1Pharmaceutical Services Department, King Hamad University Hospital, Al Sayh 24343, Bahrain or hager.salah@khuh.org.bh (H.S.); or oalsamani@uob.edu.bh (O.A.);; 2Education and Proficiency Centre, King Hamad University Hospital, Al Sayh 24343, Bahrain; 3Pharmacy Program, Allied Health Department, College of Health Sciences, University of Bahrain, Manama 32038, Bahrain; 4Cardiothoracic ICU and Anesthesia Department, Mohammed Bin Khalifa Specialist Cardiac Center, Awali 183261, Bahrain; 5Allied Health Department, College of Health Sciences, University of Bahrain, Manama 32038, Bahrain; 6Department of Clinical Pharmacy, Faculty of Pharmacy, Beni-Suef University, Beni-Suef 62521, Egypt; 7Department of Clinical Pharmacy, Faculty of Pharmacy, October 6 University, 6th October City 12511, Egypt; 8Department of Clinical Pharmacy, Faculty of Pharmacy, Nahda University (NUB), Beni-Suef 62521, Egypt; 9Department of Clinical Pharmacy, Beni-Suef University Hospitals, Faculty of Medicine, Beni-Suef University, Beni-Suef 62521, Egypt; 10Clinical Pharmacy Department, Faiyum Oncology Center, Fayium 63511, Egypt; 11Clinical Nutrition Department, Fayium Health Insurance Authority, Fayium 63511, Egypt; 12Public Health and Community Medicine Department, Faculty of Medicine, Beni-Suef University, Beni-Suef 62514, Egypt; 13Clinical Pharmacy Department, College of Pharmacy, Al Ain University, Abu Dhabi P.O. Box 112612, United Arab Emirates; 14Mental Health Department, Faculty of Education, Helwan University, Helwan 11795, Egypt

**Keywords:** COVID-19, vaccines, healthcare workers, booster doses

## Abstract

(1) Background: During 2019, the COVID-19 pandemic was threatening healthcare services and workers, and acquiring immunity was an option to stop or limit the burden of this pandemic. Herd immunity was a top priority worldwide as the virus was spreading rapidly. It was estimated that 67% of the total global population should be immunized against COVID-19 to achieve herd immunity. The aim of the current study is to investigate different perceptions of healthcare workers in the Kingdom of Bahrain and Egypt using an online survey in an attempt to evaluate their awareness and concerns regarding new variants and booster doses. (2) Methods: This study conducted a survey on healthcare workers in the Kingdom of Bahrain and Egypt about their perception and concerns on the COVID-19 vaccines. (3) Results: The study found that out of 389 healthcare workers 46.1% of the physicians were not willing to take the booster doses (*p* = 0.004). Physicians also did not support taking the COVID-19 vaccine as an annual vaccine (*p* = 0.04). Furthermore, to assess the association between the type of vaccine taken with the willingness of taking a booster vaccine, healthcare workers beliefs on vaccine effectiveness (*p* = 0.001), suspension or contact with patients (*p* = 0.000), and infection after COVID-19 vaccination (*p* = 0.016) were significant. (4) Conclusion: Knowledge about vaccine accreditation and regulation should be dispersed more widely to ensure that the population has a positive perception on vaccine safety and effectiveness.

## 1. Introduction

During 2019, the COVID-19 pandemic was threatening humanity. Acquiring immunity was an option to stop or limit the burden of this pandemic; however, the world needed 67% of the total population to be immunized against COVID-19 to reach what is known as community immunity (herd immunity) [1]. Although vaccination was prioritized for the elderly, patients with comorbidities, and those at high risk, the primary demographic intended for vaccination was healthcare workers. Healthcare workers were at the highest risk of contracting the COVID-19 infection. It was shown that 245,000 cases of COVID-19 were reported for healthcare workers in the United States for the year 2020 [2]. Furthermore, special attention was given to healthcare workers to ensure that health equity is maintained between populations [3,4].

While people all over the world were working to curb COVID-19, new strains of the virus began to emerge. The World Health Organization (WHO) identified five potentially dangerous variants: alpha, beta, gamma, delta, and omicron. Accordingly, a variant of concern is defined as one that either increases the transmissibility, causes a detrimental change in COVID-19 epidemiology, causes large change in clinical disease prevention, or decreases the effectiveness of public health and social measures that are available for diagnostics, vaccines, or therapeutics. In other words, the viral infection rates in humans increased by the presence of genes encoding for antagonists of host defense mechanisms within the viral genome, causing the emergence of COVID-19 mutations [5].

Multiple vaccines were developed by different pharmaceutical companies using different technologies. Vaccines were developed using whole viruses, RNA or mRNA, and non-replicating viral vectors. Reasons for the preference of one vaccine over others included: the reported efficacy of each vaccine, availability of vaccine in the region, and regional acceptance. In the Kingdom of Bahrain and Egypt, the Sinopharm vaccine was demanded because it was the first to be approved following a Phase III clinical trial that started in August 2020 with a request of 6000 volunteers; however, 7700 volunteers were enrolled. Later, the vaccine was granted an emergency use authorization on 3 November 2020 for frontline workers [6].

Despite receiving millions of vaccine doses in an effort to cut down on infection spread through transmission and to avoid relying solely on herd immunity, the number of cases of the disease continued to rise [7,8]. Infection cases were documented even after receiving two doses of the COVID-19 vaccine, which is known as breakthrough disease, and it had become a significant issue [9]. As a result of this phenomena, it was suggested that current vaccine strategy may not be effective in preventing the spread of the SARS-CoV-2 infection [10,11].

Controlling COVID-19 and fighting against it were associated with a number of barriers starting from isolating and studying the virus to producing vaccines and drugs which worked against this virus. Beyond the mission to find treatments for COVID-19, there were psychological factors that affected community acceptance of these treatments and included: ambiguity of information, concerns about short-term and long-term safety, anxiety of side effects, unwillingness to be vaccinated, the cost of different types of vaccines, and storage, handling, and manufacturing processes of vaccines along with monitoring of various vaccines [12,13,14]. Multiple methods were used to identify the barriers for vaccination, such as The vaccination attitudes examination (VAX) scale, depression anxiety stress scale (DASS), and other modified surveys used in cross-sectional studies [15,16,17,18,19,20].

The aim of this multicenter study is to investigate different perceptions of healthcare workers in the Kingdom of Bahrain and Egypt using an online survey in order to assess the awareness and concerns regarding new variants and booster doses.

## 2. Materials and Methods

### 2.1. Study Design

An observational, cross-sectional study was carried out using a web-based questionnaire distributed to three healthcare institutions in the Kingdom of Bahrain and Egypt from January 2022 to September 2022. Beni-Suef University, Faculty of Medicine, King Hamad University Hospital (KHUH), and Mohammed Bin Khalifa Bin Salman Al Khalifa Cardiac Centre (MKCC) participated and were included in this study. Healthcare workers from different backgrounds and specialties were included, such as physicians, pharmacists, nurses, and allied health workers (laboratory technicians, radiology technicians, and physiotherapists).

#### Inclusion and Exclusion Criteria

**Inclusion criteria:** Healthcare workers that consented to participate in the study from the three institutions.

**Exclusion criteria:** Non-healthcare workers.

### 2.2. Sample Technique

A questionnaire in the form of an online form was distributed via emails, announcements in the institutions, and shared on social media for the healthcare workers within the institutions (WhatsApp, Facebook, etc.). The questionnaire was recirculated every two weeks.

The snowballing sampling method was used in this investigation. Using Epi Info StatCalc [Info, 2014], the sample size for population survey was calculated at 95% confidence level, 5% acceptable margin of error, 1 design effect, 50% expected frequency (of regular follow-up or positive DASS), and the minimum sample size was found to be at least 334 persons.

### 2.3. Data Collection Tool

A pilot study was done on thirty participants prior to the actual data collection to validate the questionnaire’s readability and usability that was created by the authors in English. The validity was evaluated using Cronbach’s alpha, and it was found that all domains of the questionnaire had an adequate level of validity (Cronbach’s alpha value of 0.6 or higher), with the exception of the perceived barrier, which had a value of 0.59.

Part 1: A 10-item self-structured section that evaluated the socio-demographic data of participants including: age, gender, weight (kg), height (cm), residence, occupation, seniority level, experience, health state, allergies (if any), and smoking

Part 2: A 22-item self-administered section asking about mandatory vaccines, status of COVID-19 vaccination, infection with COVID-19 after vaccination, times of getting infection after vaccination, if the healthcare workers are confident about the COVID-19 vaccine, if they believe that the COVID-19 vaccine is safe or effective, vaccine type that they were administered, if their institutions were handling COVID-19 patients (including the vaccination process), about experience with any side effects after vaccination, information sources of healthcare workers, and what concerns they have regarding the COVID-19 vaccine, their perception of COVID-19 vaccine, if the booster dose is effective against new COVID-19 mutations, and about the effectiveness of administration of mixed vaccine brands.

Part 3: A 8-item self-administered section adapted from the COVID-19 vaccine attitude examination scale.

### 2.4. Ethical Consideration

The participation of any individual in the study was entirely voluntary, and ethical approval was obtained from each and every institute before the study was carried out. Every piece of information was kept anonymous and strictly confidential. The study was in accordance with the Declaration of Helsinki, and ethical approvals were obtained from the Institutional Review Board (or Ethics Committee) of Beni-Suef University, Faculty of Medicine, King Hamad University Hospital, and Mohammed Bin Khalifa Specialist Cardiac Center with registration numbers (BSU) (FWA00015574), (KHUH) (IRB #22-497), and (MKCC) (2022-0071), respectively.

### 2.5. Statistical Analyses

The Statistical Package for Social Science (SPSS) version 25 was used to gather, code, and analyze data (IBM, USA). We estimated the frequency distribution, percentage, and descriptive statistics. The independent student T test was used to discover the difference between the two categories, and the Chi-Square Test of Independence was utilized to determine a connection between categorical variables. Binary logistic regression was used to identify the determinants of no follow-up. *p* values of ≤ 0.05 were considered significant.

## 3. Results

A total of 389 healthcare workers consented to participate from Egypt (41.6%) and Bahrain (58.4%). There were more females (65.6%) than males (34.4%) and majority of the population was from pharmacy background (35%), followed by nurses (31.6%), physicians (22.9%), and allied health personnels (10.5%), as shown in Figure 1.

Over half of the population were of a higher seniority level with experience of more than 10 years (59.1%) and those of a lower seniority level in their occupation accounted for less than half of the population (40.7%). Majority of the population was healthy (79.9%) and the rest (20.1%) had comorbidities. A similar trend followed with a majority of the population being non-smokers (88.9%) and did not have any allergies (79.9%). This trend continued when participants were asked if they follow their mandatory vaccine regimen, and 88.4% of the population answered yes. Similarly, 96.7% had been vaccinated against COVID-19.

The population was split in half about infection of COVID-19 after vaccination. Although half of the population (48.1%) believe that more data is required about the safety of COVID-19 vaccines, more than half of the population was confident to receive COVID-19 vaccination (60%). Around 31% of the participants took the vaccine as soon as it was launched (31.6%), as shown in (Table 1).

Perceptions of healthcare workers on COVID-19 vaccinations are shown in Table 1. The awareness of precautionary measures against the spread of the virus was widespread, and this was proven as 79.2% of respondents were educated on the matter during the COVID-19 pandemic. Most of the healthcare workers had strong opinions about the safety and efficacy of the COVID-19 vaccines, and majority answered “certain brands” or “more information required” in their beliefs. This was also seen when healthcare workers were asked what was their concerns in regards to the COVID-19 vaccine, and majority responded “safety” and “efficacy”. Some of the respondents also highlighted the fact that “emergency approvals” were a point of concern. They had a similar opinion when asked about the administration of mixed vaccine brands.

Most questions received a positive response except for when healthcare workers were asked about COVID-19 becoming an annual vaccine. An in-depth analysis was conducted based on occupations due to the insignificance of the question. This showed that healthcare workers that were not in support of taking the COVID-19 vaccine as an annual vaccine were 46.1% of physicians, 26.0% of nurses, and 34.6% of pharmacists who answered “no” for that question (*p* = 0.04). This may be due to the fact that physicians (46.1%) were not willing to take the booster (*p* = 0.004), pharmacists thought that the booster dose was not effective against the COVID-19 mutations (28.4%) (*p* = 0.001), and nurses assumed that administration of mixed vaccine brands was not a safe practice (*p* = 0.04).

The top three types of COVID-19 vaccines taken by healthcare workers were Pfizer–BioNTech, Sinopharm, and AstraZeneca–Oxford (Figure 2). The reported side effects are illustrated in Figure 3, and fatigue and fever were reported in over 50% of the cases.

Univariate regression was performed to assess the association between the type of vaccine taken with the side effects and occurrence of infection after vaccination. Results showed that side effects after vaccination (*p* = 0.000), belief in vaccine effectiveness (*p* = 0.001), suspension or contact with patients (*p* = 0.000), and infection after COVID-19 vaccination (*p* = 0.016) were significant. Univariate regression was also performed to assess the association of demographic variables with booster acceptance among the vaccinated population. Results showed that gender (*p* = 0.023) and participants’ country of residence (*p* = 0.002) were significant.

Furthermore, multivariate regression was conducted to further determine the factors associated with booster willingness. Those who were not following their mandatory vaccines regimen (*p* < 0.001) did not believe that a booster dose was effective against new COVID-19 mutations (*p* = 0.041), and they being male (*p* = 0.038), and having a normal BMI (*p* = 0.027) showed significant association with booster willingness. Additionally, bivariate analysis was performed to calculate the factors affecting the feeling of vaccine safety (Table 2). The feeling of vaccine safety was significant for those that agreed (*p* = 0.002) and was also associated with neutral (*p* = 0.001) and negative feelings (*p* = 0.006). The feeling of dependence on the vaccine to eliminate COVID-19 severity was significant for those that strongly agreed (*p* < 0.001), agreed (*p* < 0.001), or were neutral (*p* < 0.001).

It is important to discuss both the feelings and perceptions of healthcare workers in the study to understand their perspective and concerns towards the COVID-19 vaccines. It has been noted from both results that vaccine hesitancy was high due to the lack of information, feelings that the emergency approval was not necessary, and the perception developed from the lack of information on the safety and efficacy of the COVID-19 vaccines. These issues proved to be significant throughout the assessment.

## 4. Discussion

Vaccination hesitancy persists as the COVID-19 pandemic progresses despite the fact that vaccines have received full approval for use in several countries and millions of doses have been administered globally [21]. As the pandemic progresses, the debate has centered on employers’ and governments’ roles in encouraging, if not mandating, vaccination. Discussions about these ideas have intersected with various political and philosophical ideologies, resulting in debate due to the strong opinions on both sides [22].

The current study showed that safety and efficacy of the vaccine among healthcare workers was a major concern that required more data. However, respondents were more confident of effectiveness where majority of responses showed that certain brands were more effective than others. According to one recent study, it was demonstrated that trust must be built and education regarding the vaccines should be spread to reduce the vaccination hesitancy [23,24]. The lack of trust in this period of the pandemic was due to the lack of long-term data about vaccine and emergency approvals for vaccines as well as minimal data about safety and effectiveness. Concern about safety and efficacy is one of the most common reasons for COVID-19 vaccine hesitancy [25,26,27]. Some of these reservations are related to the emergency approval given by FDA to vaccines [28]. Emergency FDA approval is a procedure that was previously used in public health emergencies, and the emergency-approved vaccines later received proper approval by the FDA in August 2021 (Pfizer) and January 2022 (Moderna) [29,30]. Therefore, many healthcare workers delayed receiving or making a decision on the COVID-19 vaccine until sufficient information about its safety and efficacy became available [31]. 

In our study, majority of physicians were not willing to take the booster doses as compared to other healthcare workers. Physicians were also not in support of taking COVID-19 as an annual vaccine in contrast to other studies [25,26,28,32,33,34]; however, other studies suggested that there was no association between willingness to get a booster dose and being a physician [35]. This may largely be due to the practice of evidence-based medicine. Awareness of the COVID-19 infection and vaccines was significantly linked to vaccination acceptance (*p* = 0.00) [36,37], which proved to be in line with our findings that most healthcare workers were vaccinated as soon as vaccines were rolled out. With a higher level of education about precautionary measures, healthcare workers are more likely to be vaccinated with booster doses or annual shots due to the fact that information about vaccine safety and efficacy would be easily accessible. This may be attributed to a higher clinical research experience in some professions.

Unlike other studies, majority of the participants in this study registered to have the vaccine as soon as it is announced, and majority of them had the choice of selecting between different vaccines, whereas other studies reported difficulty in receiving vaccination [38]. This could largely be due to the fact that healthcare workers were not employed by a major hospital or healthcare organization. Additionally, some may have lived at a significant distance from vaccination sites during the early phase of vaccine roll out when distribution was limited. Financial barriers to receiving the vaccine were unlikely in areas where the vaccine was widely available and employers would allow healthcare workers to be vaccinated [39].

Aside from the western world, studies that focused on vaccine hesitancy in the Middle East and North African region were conducted in Kuwait and Ethiopia [40]. These studies were conducted using a similar online format as used in our study. In Kuwait, it was reported that despite high vaccine acceptability, healthcare workers did indicate their unwillingness to vaccinate, which may be due to the low to intermediate levels of general negative attitude towards vaccination [40]. These studies, similar to ours, suggested that vaccine hesitancy was dependent on the depth of information that was provided by health authorities.

Studies in Saudi Arabia about COVID-19 perceptions, acceptance, and hesitancy were the most common in the region, which may be due to the large area of the country. Unlike other studies published, some were published prior to the arrival of vaccines [41,42,43,44,45,46,47]. These studies concluded, similar to our study, there was an association between vaccine willingness and concerns about vaccine safety and efficacy, i.e., beliefs that vaccines may not be sufficiently tested. Furthermore, they mentioned that there was a mistrust in pharmaceutical companies, which might have been due to the fact that COVID-19 vaccination received an emergency FDA approval [29,30].

A systematic review conducted on the seven regions of the world showed that the Middle East and the North African region had the lowest percentage of studies conducted on COVID-19 vaccine acceptance [48]. This study highlighted that vaccine hesitancy was noticed to be the lowest in that region; however, it identified a worldwide gap pertaining to the lack of knowledge on vaccination acceptance. Interestingly, it was reported that a higher level of vaccination literacy may be a requirement for a widespread vaccination coverage due to its association with a higher level of knowledge [49].

A systematic review that included 49 published papers from 2020 to 2022 in the Gulf Cooperation Countries and the Middle Eastern countries broadly highlighted vaccine hesitancy [50]. The study highlighted that there were no studies that focused on vaccination hesitancy in the Kingdom of Bahrain and Egypt. Thus, this manuscript, as per our knowledge, is the first report on vaccination hesitancy in Bahrain.

The main strength of this study is that there was no standardized protocol for the COVID-19 pandemic control. Each country depended on the amount of resources that they had, and the primary sources of information were the WHO and the CDC. Due to the multi-centric approach of this study, we were able to assess the willingness of booster doses between healthcare workers, thus, eliminating the fact that protocols and lockdowns were different in various countries. Similar to other studies, the current study confirmed that information regarding the safety and efficacy of vaccines as well as their approval processes should be incorporated by health authorities to design effective interventions for healthcare workers and for the general public. The importance of knowledge and awareness of the COVID-19 vaccines was reported to increase the acceptance rate of their administration [51]. In addition, the need for the design of effective educational interventions to increase the receipt of the COVID-19 vaccine has been documented [40].

Moreover, another study conducted on non-healthcare workers reported that participants who had high knowledge scores (91%) were significantly more likely to be vaccinated compared to those who scored low knowledge scores. This finding shed light on the effect of type and quality of information and professional background on the vaccine acceptance rate [51].

The limitations of this study include is that it was as an online survey taken by healthcare workers that consented to becoming a part of the study. The nature of the online study did not allow healthcare workers to express their beliefs appropriately as they may have had different views on the matter. In addition, the limitation of answering the questions could have caused misunderstandings due to the multiple choice only format. Furthermore, use of technology by healthcare workers prior to the COVID-19 pandemic could have been limited, reducing our sample size. For future research, we recommend that researchers dive into understanding the beliefs and perceptions of healthcare workers regarding the up-to-date accreditation and regulation of vaccines and to understand the gap in medical education. Future research is needed regarding vaccine literacy programs that can aid in future health emergencies and decrease the rates of vaccine hesitancy.

## 5. Conclusions

The current study concluded that there is an urgent need for effective knowledge accessibility regarding COVID-19 vaccines in order to face future pandemics.

In addition, it has been concluded that there was a low willingness among healthcare workers to take the booster dose and an even lower willingness to take the COVID-19 vaccine as an annual vaccine. As a solution, we recommend a structured program that may increase the vaccine literacy in the healthcare community.

## Figures and Tables

**Figure 1 vaccines-11-01061-f001:**
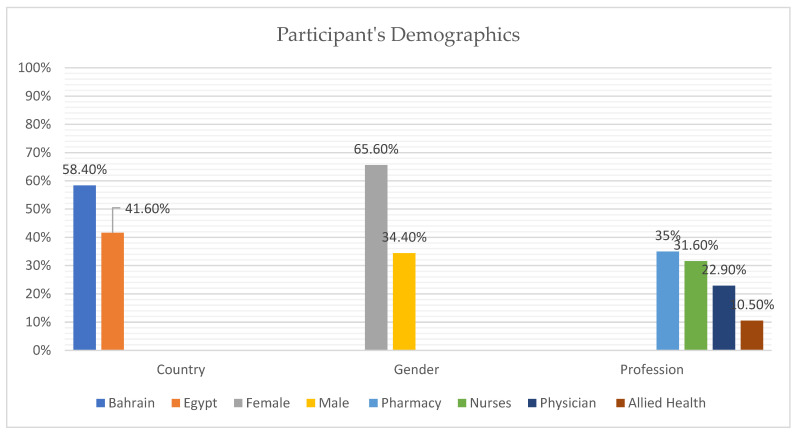
Participant Demographics Data.

**Figure 2 vaccines-11-01061-f002:**
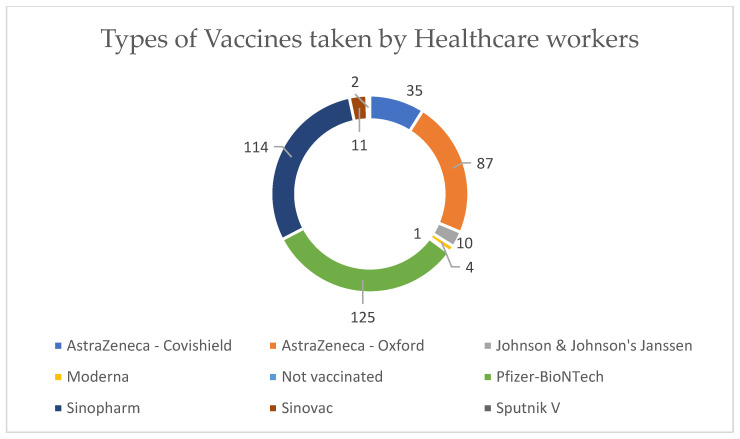
Types of vaccines taken by healthcare workers.

**Figure 3 vaccines-11-01061-f003:**
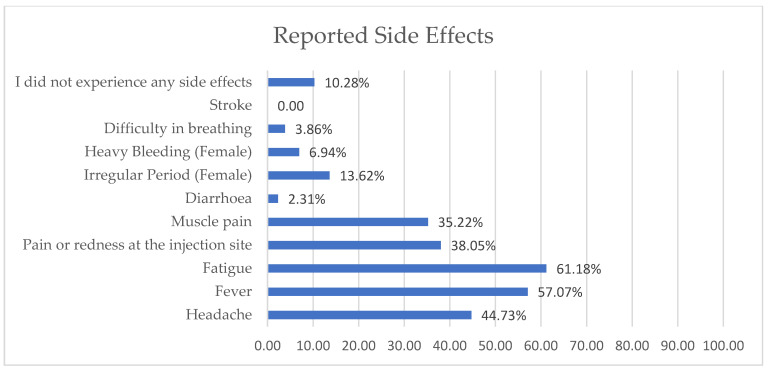
Side effects reported by healthcare workers.

**Table 1 vaccines-11-01061-t001:** Perceptions of healthcare workers on the COVID-19 vaccines.

Questions	Responses (%)	*p*-Value
Do you believe that COVID-19 vaccine is safe?		*p* < 0.001
Yes	92 (23.7%)	
No	45 (11.6%)	
Certain brands	65 (16.7%)	
More data required	187 (48.1%)	
Do you believe that COVID-19 vaccine is effective?		*p* < 0.001
Yes	103 (26.5%)	
No	40 (10.3%)	
Certain brands	133 (34.2%)	
More data required	113 (29.0%)	
If yes, after how long of vaccine availability did you register?		*p* < 0.001
Once its announced	123 (31.6%)	
Within 1 month	66 (17.0%)	
1–3 months	73 (18.8%)	
4–6 months	47 (12.1%)	
More than 6 months	80 (20.6%)	
Did you get the option to select between vaccines?		0.002
Yes	225 (57.8%)	
No	164 (42.2%)	
Is your institution handling COVID-19 patients (Including vaccination process)?		*p* < 0.001
Yes	272 (69.9%)	
No	117 (30.1%)	
What concerns regarding COVID-19 vaccine do you have? *		*p* < 0.001
Safety	255 (65.5%)	
Efficacy	215 (55.2%)	
No Concerns	61 (15.6%)	
New technology	51 (13.1%)	
Emergency approval	145 (37.2%)	
What is your perception about COVID-19 vaccine? *		*p* < 0.001
Expected not make any difference	25 (6.4%)	
Expected reduce infection severity	269 (69.15%)	
Expected reduce the morbidity and mortality	257 (66.06%)	
Expected reduce the pandemic spread	176 (45.24%)	
From your practice with suspected/confirmed COVID-19 patients, what did you feel?		*p* < 0.001
I am not in contact with suspected/confirmed COVID-19 patients.	104 (26.7%)	
Vaccinated and non-vaccinated are sharing the same risk of morbidity and mortality.	94 (24.2%)	
Vaccinated patients are at high protection rate.	191 (49.1%)	
If COVID-19 vaccine become an annual vaccine, are you willing to take it?		0.349
Yes	117 (30.1%)	
No	132 (33.9%)	
Not sure	140 (36.0%)	
If your vaccine brand required a booster dose, are you willing to take it?		*p* < 0.001
Yes	169 (43.4%)	
No	120 (30.8%)	
Not sure	100 (25.7%)	
Do you believe that booster dose is effective against new COVID-19 mutations?		*p* < 0.001
Yes	139 (35.7%)	
No	61 (15.7%)	
Not sure	189 (48.6%)	
Do you believe that mixing between vaccine brands is a safe practice?		*p* < 0.001
Yes	56 (14.4%)	
No	103 (26.5%)	
Not sure	230 (59.1%)	
Were there campaigns, lectures or educational sessions discussing COVID vaccine and the public concerns at your organization?		*p* < 0.001
Yes	308 (79.2%)	
No	81 (20.8%)	

* participants can choose more than one option.

**Table 2 vaccines-11-01061-t002:** Feelings of healthcare workers towards the safety of COVID-19 vaccines.

Questions	Responses	*p*-Value
I feel safe after being vaccinated		*p* < 0.001
Strongly agree	37 (9.5%)	
Agree	97 (24.9%)	
Neutral	146 (37.5%)	
Disagree	74 (19.0%)	
Strongly disagree	35 (9.0%)	
I can depend on the vaccine to eliminate COVID-19 severity		*p* < 0.001
Strongly agree	41 (10.5%)	
Agree	121 (31.1%)	
Neutral	121 (31.1%)	
Disagree	50 (12.9%)	
Strongly disagree	56 (14.4%)	
Vaccine are safe for child and pregnant ladies		*p* < 0.001
Strongly agree	24 (6.2%)	
Agree	40 (10.3%)	
Neutral	113 (29.0%)	
Disagree	74 (19.0%)	
Strongly disagree	138 (35.5%)	
Natural Immunity would be more safe and effective to stop COVID-19		*p* < 0.001
Strongly agree	62 (15.9%)	
Agree	72 (18.5%)	
Neutral	177 (45.5%)	
Disagree	42 (10.8%)	
Strongly disagree	36 (9.3%)	
The COVID-19 vaccine might have long term side effects		*p* < 0.001
Strongly agree	60 (15.4%)	
Agree	75 (19.3%)	
Neutral	174 (44.7%)	
Disagree	57 (14.7%)	
Strongly disagree	23 (5.9%)	
Emergency approvals for COVID-19 vaccines was not necessary		*p* < 0.001
Strongly agree	43 (11.1%)	
Agree	35 (9.0%)	
Neutral	113 (29.0%)	
Disagree	102 (26.2%)	
Strongly disagree	96 (24.7%)	
Data regarding COVID-19 is published transparently		*p* < 0.001
Strongly agree	40 (10.3%)	
Agree	65 (16.7%)	
Neutral	174 (44.7%)	
Disagree	78 (20.1%)	
Strongly disagree	32 (8.2%)	
Are you confident to COVID-19 vaccine?		*p* < 0.001
Strongly agree	38 (7%)	
Agree	62 (13%)	
Neutral	159 (20%)	
Disagree	79 (27%)	
Strongly disagree	51 (33%)	

## Data Availability

The data will be available upon request from the corresponding authors.

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
