# Peer review of "COVID-19 Booster Doses: A Multi-Center Study Reflecting Healthcare Providers’ Perceptions"

_vaccines, 2023, doi:10.3390/vaccines11061061_

Round 1

Reviewer 1 Report

Thank you for the chance to review this manuscript. Salah et al. studied the perception of healthcare providers regarding COVID-19 vaccine booster doses. They reported a low intention of healthcare workers to take the booster. This is interesting results for the scientific community.

Please see below some major and minor concerns.

Major:

1- The manuscript needs full English editing. There are unclear sentences and grammatical errors.

2- Please explain how the survey questions were developed? have they been adopted from a published survey?  what language was used?

4- Line 127 indicate that validity was evaluated. How this was done? By whom? also what was tested; content or construct validity? As this survey was not published before, these information is important. Unless the authors used a previously published and validated survey.

5-  Please provide the survey questions as supplementary data.

6-  Please provide a paragraph about the limitation of the study.

Minor:

1- Abstract: please add a sentence about the aim of the study

2- it would be better to use the term "healthcare workers" throughout the manuscript. Term "healthcare providers" may indicate the institutes and hospital as a  whole more than individuals and this may confuse some readers.  

2- Discussion needs some improvement. Author discussed their results with other healthcare worker results which is good. However, it would be nice to compare the results with other groups like students, refugees, elderly....etc from the same geographical region. please see below some references as examples:

DOI: 10.3390/vaccines10101634

  •  
  • DOI: 10.3390/ijerph18168836

manuscript needs full English editing.

Author Response

Comment

Author Response

Text Insertion (if applicable)/ section

Major:

1- The manuscript needs full English editing. There are unclear sentences and grammatical errors.

2- Please explain how the survey questions were developed? have they been adopted from a published survey?  what language was used?

4- Line 127 indicate that validity was evaluated. How this was done? By whom? also what was tested; content or construct validity? As this survey was not published before, these information is important. Unless the authors used a previously published and validated survey.

5-  Please provide the survey questions as supplementary data.

6-  Please provide a paragraph about the limitation of the study.

Minor:

1- Abstract: please add a sentence about the aim of the study

2- it would be better to use the term "healthcare workers" throughout the manuscript. Term "healthcare providers" may indicate the institutes and hospital as a  whole more than individuals and this may confuse some readers.  

2- Discussion needs some improvement. Author discussed their results with other healthcare worker results which is good. However, it would be nice to compare the results with other groups like students, refugees, elderly....etc from the same geographical region. please see below some references as examples:

DOI: 10.3390/vaccines10101634

·          

·         DOI: 10.3390/ijerph18168836

Comments on the Quality of English Language

manuscript needs full English editing.

The manuscript was proofread by a native speaker in King Hamad University Hospital that has proofreading credentials.

The survey was mixed between author created questions and questions adapted from the COVID-19 Vaccine Attitude Examination Scale. The survey was only distributed in English. You can find it in detail under section 2.3 Data collection Tool. It was edited to allude to this point.

We appreciate your comment. The authors conducted statistical analysis using SPSS v.25. The testing used was added to the sentence on Line 127.

Attached.

A paragraph highlighting limitations was added prior to conclusion.

Done

Done

Israa

The manuscript was proofread by a native speaker in King Hamad University Hospital that has proofreading credentials.

Reviewer 2 Report

Comments

I HAVE MADE EDITORIAL AND OTHER SUGGESTIONS FOR THE FIRST 100 LINES  … THIS SAME EFFORT IS NEEDED THROUGHOUT THE PAPER

Line 35  - should use the Oxford comma for better reading .. “and providers, and acquiring”

Line 40 &41 does not make sense … as compared … to what?

Line 50  Rather than a comma, a full stop should be used before “Acquiring”

Line 54 Remove comma  … “comorbidities, and”

Line 59 Rather than “has been put into action” … was put into action by ??? (governments in Bahrain and Egypt??? Please clarify

Line 61 & 62 Should read …  Additionally, special attention was given to healthcare providers in order to ensure health equity between populations

## NOTE: The content in the paragraph introduces the paper well.

Line 65  After omicron (note space before full stop .. add “at that time  - or be specific (month/date)

Line 65 – 68 Accordingly, a variant of concern is defined as either increasing the transmissibility, causing a detrimental change in COVID-19 epidemiology, increasing change in clinical disease prevention, or decreasing the effectiveness of public health and social measures available for diagnostics, vaccines, or therapeutics. In other words,

Line 70 – 72  a result, increased rates of vaccination lead to an increase in the number of newly emerged virus strains, such as delta and omicron variants(5).

## NOTE THIS LINE IS OF SERIOUS CONCERN. In my mind, this is a misinterpretation of the Gomez paper. The sentence does not follow from the previous parts of the paragraph .. perhaps it is just badly worded

Line 87     and it had become a significant issue   

Line 87 Space after full stop

Line 90 & 91 Controlling  COVID-19 and fighting against it were associated with a number of barriers, starting from isolating and studying the virus to produce vaccines and drug which work against this virus.

Line 92-95 Moreover, beside the mission to find treatments for COVID 19 there were psychological factors that affected community acceptance for this treatment. This included: ambiguity of information; concerns about short-term and long term safety; anxiety from side effects; unwillingness to be vaccinated; the costs of different types of vaccines; storage conditions; handling; manufacturing methods; and monitoring of various vaccines (12–14).

## NOTE This study does need to have the data collection and discussion put in a time context

Materials and Methodology (SECTION 2)

English usage needs as much work as in the first section.

This section could be improved by noting the weaknesses of the study  - although they are, in effect, stated in the paper. For example, this was 1)self-selecting, 2) online study 3) understanding the number of people who took part in the study compared to the number of healthcare workers in the three hospitals and specifically, the number of physicians.

Although a “snowball” technique – I was interested to understand how much follow up was necessary to engage the number of people from the hospitals.

Results Section

English usage needs as much work as in the first section.

Lines 218-225 appear to argue the opposite to what the figures are showing (where the % are not left out for particular professions.

Majority of the physicians [41 (46.1%)][M1]  were not willing to take the booster doses as compared to pharmacists [42 (30.9%)[M2] ] and nurses 27 (22.0%) [ p=0.004]. Physicians are also not in the support of taking COVID -19 as an annual vaccine [41 (46.1% vs nurses [32 (26.0%] and pharmacists [47 (34.6%)], (p= 0.04). Majority of the pharmacists think that the booster dose is not effective against the COVID-19 mutations (28.4%) vs physicians [8(9.0%)] and nurses [15 (12.2%)] [p=0.001]. Majority of the nurses think that mixing vaccine brands is not a safe practice [39 (31.7%) vs pharmacist [35 (25.7%)] and physicians [13 (14.6%)]. [p= 0.04[M3] ].

It is a similar situation in the next paragraph

 [M1]This is less than a majority

 [M2]This is less than a majority and less than physicians.

 [M3]Maybe the percentages are back the front . it certainly seems so from the words.

The English language is satisfactory

Author Response

Comment

Author Response

Text Insertion (if applicable)/ section

I HAVE MADE EDITORIAL AND OTHER SUGGESTIONS FOR THE FIRST 100 LINES  … THIS SAME EFFORT IS NEEDED THROUGHOUT THE PAPER

Line 35  - should use the Oxford comma for better reading .. “and providers, and acquiring”

Line 40 &41 does not make sense … as compared … to what?

Line 50  Rather than a comma, a full stop should be used before “Acquiring”

Line 54 Remove comma  … “comorbidities, and”

Line 59 Rather than “has been put into action” … was put into action by ??? (governments in Bahrain and Egypt??? Please clarify

Line 61 & 62 Should read …  Additionally, special attention was given to healthcare workers in order to ensure health equity between populations

## NOTE: The content in the paragraph introduces the paper well.

Line 65  After omicron (note space before full stop .. add “at that time  - or be specific (month/date)

Line 65 – 68: Accordingly, a variant of concern is defined as either increasing the transmissibility, causing a detrimental change in COVID-19 epidemiology, increasing change in clinical disease prevention, or decreasing the effectiveness of public health and social measures available for diagnostics, vaccines, or therapeutics. In other words,

Line 70 – 72  a result, increased rates of vaccination lead to an increase in the number of newly emerged virus strains, such as delta and omicron variants(5).

## NOTE THIS LINE IS OF SERIOUS CONCERN. In my mind, this is a misinterpretation of the Gomez paper. The sentence does not follow from the previous parts of the paragraph. perhaps it is just badly worded

Line 87     and it had become a significant issue   

Line 87 Space after full stop

Line 90 & 91 Controlling COVID-19 and fighting against it were associated with a number of barriers, starting from isolating and studying the virus to produce vaccines and drug which work against this virus.

Line 92-95 Moreover, beside the mission to find treatments for COVID 19 there were psychological factors that affected community acceptance for this treatment. This included: ambiguity of information; concerns about short-term and long term safety; anxiety from side effects; unwillingness to be vaccinated; the costs of different types of vaccines; storage conditions; handling; manufacturing methods; and monitoring of various vaccines (12–14).

## NOTE This study does need to have the data collection and discussion put in a time context

Materials and Methodology (SECTION 2)

English usage needs as much work as in the first section.

This section could be improved by noting the weaknesses of the study  - although they are, in effect, stated in the paper. For example, this was 1)self-selecting, 2) online study 3) understanding the number of people who took part in the study compared to the number of healthcare workers in the three hospitals and specifically, the number of physicians.

Although a “snowball” technique – I was interested to understand how much follow up was necessary to engage the number of people from the hospitals.

Results Section

English usage needs as much work as in the first section.

Lines 218-225 appear to argue the opposite to what the figures are showing (where the % are not left out for particular professions.

Majority of the physicians [41 (46.1%)][M1]  were not willing to take the booster doses as compared to pharmacists [42 (30.9%)[M2] ] and nurses 27 (22.0%) [ p=0.004]. Physicians are also not in the support of taking COVID -19 as an annual vaccine [41 (46.1% vs nurses [32 (26.0%] and pharmacists [47 (34.6%)], (p= 0.04). Majority of the pharmacists think that the booster dose is not effective against the COVID-19 mutations (28.4%) vs physicians [8(9.0%)] and nurses [15 (12.2%)] [p=0.001]. Majority of the nurses think that mixing vaccine brands is not a safe practice [39 (31.7%) vs pharmacist [35 (25.7%)] and physicians [13 (14.6%)]. [p= 0.04[M3] ].

It is a similar situation in the next paragraph

 [M1]This is less than a majority

 [M2]This is less than a majority and less than physicians.

 [M3]Maybe the percentages are back the front . it certainly seems so from the words.

Done

Removed

Done

Done

Removed

Thank you.

Done

Done

Rephrased.

Done

Done 

Done

Done

Done

Done

This has been included in the limitations.

This was included in section 2.3 Data Collection

Done

We have looked over these paragraphs and as part of the editing and proofreading. We have adjusted the paragraph  to adhere to the figures and follow parallelism. 

Reviewer 3 Report

This paper is hard to follow & needs the assistance of a professional medical writer. 

Abstract:

Change "acquiring immunity" and "herd immunity" to "vaccination" and "high vaccination coverage", respectively.

Results -  Incomplete sentence in first line physicians not willing to take the booster as compared to who?

Conclusion: Accreditation is not the right word.  Assume you are referring to licensure process and data needed to do so.

BACKGROUND:

Same edits re acquired immunity as noted above.

Repeated word "providers" in line 61

Discuss when vaccines first available in the countries studied & how distributed to HCPs.

Line 127 - Explain what is meant by "content is accurate" (line 127).   Do you mean the survey was well received or understood?

Line 133 - was any information collected on family size or living arrangements?  If not, this should be mentioned as a limitation as these variables may influence desire to be vaccinated

Line 145 - incomplete sentence - "A question regarding the mixing between vaccine brands"

Line 147 - briefly describe questions adapted from the COVID-19 Vaccine Attitute Examination Scale

Line 175 - describe allied health personnel

Line 199- define senior vs junior seniority - how many years of experience?

Line 208 - 96.75 had been vaccinated with how many doses?1 or more? 2 or more ?  Please describe. 

Table 1 - many questions seem to be missing the response of "I don't know" or "Other"  and the question about vaccine requiring a booster should have had the response of "booster not required".  Where these options available as survey responses?  If not, should clarify that this was a limitation of your survey methods.

Line 257 - Odds ratios are mentioned, but do not see them in the tables or text.  Please add.

Lines 262-268 - this text is very confusing and needs to be simplified and written more clearly. 

Table 2 - data as presented do not look like a typical multivariate analysis but only a bivariate analysis.  Please clarify.

Line 275 - change "vaccine suspicion" to "vaccine hesitancy" throughout the text 

Line 287 - begin with "lack of" long-term data etc. 

Line 296 - This is a Discussion section and it would help to discuss why physicians are not supporting of booster doses.  Please expand. 

Line 299-301 - this sentence is unclear & needs to be rewritten

Line 311 - disagree that other studies reported difdiculty in receiving vaccines.  See publications by Judith McKenzie at the Hospital of Pennsylvania.

Line 313 - please describe how and where vaccines were distributed.  Wan't vaccine available at work?

Line 327-329 - discussion of vaccination of children seems out of place.  Please explain connection to this study.

Line 338-339 - you conculde that more information are needed regarding accreditation and regulation of vaccines.  Was this specifically asked in the survey?  It could also be an access problem or simply education about the vaccine.  Suggest reconsider your conclusion.  

See my note to the editor.  Very hard to read this paper.  Needs extensive rewriting with a medical editor.

Author Response

Comment

Author Response

Text Insertion (if applicable)/ section

BACKGROUND:

Same edits re acquired immunity as noted above.

Repeated word "providers" in line 61

Discuss when vaccines first available in the countries studied & how distributed to HCPs.

Line 127 - Explain what is meant by "content is accurate" (line 127).   Do you mean the survey was well received or understood?

Line 133 - was any information collected on family size or living arrangements?  If not, this should be mentioned as a limitation as these variables may influence desire to be vaccinated

Line 145 - incomplete sentence - "A question regarding the mixing between vaccine brands"

Line 147 - briefly describe questions adapted from the COVID-19 Vaccine Attitude Examination Scale

Line 175 - describe allied health personnel

Line 199- define senior vs junior seniority - how many years of experience?

Line 208 - 96.75 had been vaccinated with how many doses?1 or more? 2 or more ?  Please describe. 

Table 1 - many questions seem to be missing the response of "I don't know" or "Other"  and the question about vaccine requiring a booster should have had the response of "booster not required".  Where these options available as survey responses?  If not, should clarify that this was a limitation of your survey methods.

Line 257 - Odds ratios are mentioned, but do not see them in the tables or text.  Please add.

Lines 262-268 - this text is very confusing and needs to be simplified and written more clearly. 

Table 2 - data as presented do not look like a typical multivariate analysis but only a bivariate analysis.  Please clarify.

Line 275 - change "vaccine suspicion" to "vaccine hesitancy" throughout the text 

Line 287 - begin with "lack of" long-term data etc. 

Line 296 - This is a Discussion section and it would help to discuss why physicians are not supporting of booster doses.  Please expand. 

Line 299-301 - this sentence is unclear & needs to be rewritten

Line 311 - disagree that other studies reported difficulty in receiving vaccines.  See publications by Judith McKenzie at the Hospital of Pennsylvania.

Line 313 - please describe how and where vaccines were distributed.  Wasn't vaccine available at work?

Line 327-329 - discussion of vaccination of children seems out of place.  Please explain connection to this study.

Line 338-339 - you conclude that more information are needed regarding accreditation and regulation of vaccines.  Was this specifically asked in the survey?  It could also be an access problem or simply education about the vaccine.  Suggest reconsider your conclusion.  

Comments on the Quality of English Language

See my note to the editor.  Very hard to read this paper.  Needs extensive rewriting with a medical editor.

The manuscript was proofread by a native speaker in King Hamad University Hospital that has proofreading credentials.

Done

Our manuscript/research does not touch this point as we are assessing the attitudes towards vaccinations in the countries. We aimed to do this study after distribution as we asked questions in regards to safety and effectiveness.

 Removed Sentence.

Included in Limitations.

Edited

Questionnaire was cited and included in the supplementary material.

Included in section 2.1 Study design

Included

There is no mention of 96.75% in the manuscript.

Yes, these options were available. The survey is attached as a supplementary material.

Removed from the manuscript.

Done

Edited.

Done

Done

Done

Done

Done.

As this is a healthcare workers based study in two countries. We did not mention the location of vaccination as it may have been different.

Removed.

Done

Round 2

Reviewer 1 Report

Thanks for the corrections which have been done so far. However, authors did not respond to come of my comments correctly.

Major point 2, based on author's answer, the survey needs to be validated. 

Major point 3, I was not asking about the software, actually SPSS does not do validity evaluation. Authors need to understand the comment before providing the answer. Please re-read the comment and respond accordingly.

Minor point 2, author's response was "Israa"!! what does that mean?? Also, I could not find that the discussion was updated with the raised points and references. Please read the comment and respond accordingly.  

Author Response

Comment

Author Response

Major point 2, based on author's answer, the survey needs to be validated.

Major point 3, I was not asking about the software, actually SPSS does accordingly. not do validity evaluation. Authors need to understand the comment before providing the answer. Please re-read the comment and respond

Minor:

Minor point 2, author's response was "Israa"!! what does that mean??

Also, I could not find that the discussion was updated with the raised points and references. Please read the comment and respond accordingly. 

Thanks for your comment,

This survey consists of three parts, part one and two self-structures which were validated internally through the pilot test of 15 responses, the validity was evaluated using Cronbach's alpha, and it was found that all domains of the questionnaire had an adequate level of validity. Furthermore, we depend also on content validity (from literature) and criterion validity (correlations).

The third part was adapted from Vaccination Attitudes Examination (VAX) Scale https://www.vax-scale.com/services/.

We apologize for the confusion , Dr.Israa (an author  of the current manuscript) is a native speaker (Canadian citizen) who conducted language editing for the whole manuscript

We apologize for the misunderstanding ,

The discussion was updated according to your appreciated comment

Reviewer 3 Report

Still not clear why responses to survey questions such as "Don't know" or "other" not shown in Table 2.

Would be useful to know the vaccine coverage rate (1, 2, 3 doses etc.) for this population.

Improved writing, but still need someone to reread this paper who is highly skilled in English.  There are typos (no "e" in awareness on line 209), use of inappropriate words ("insignificance" doesn't make sense on line 220, "preceptions", not "perceptions" on line 208), words missing (line 331), etc.   

Title does not make sense - please remove " variants of concern against".

Title should say multicenter, not multicenters 

Please review the entire manuscript again for errors such as these 

Author Response

Comment

Author Response

Still not clear why responses to survey questions such as "Don't know" or "other" not shown in Table 2.

Would be useful to know the vaccine coverage rate (1, 2, 3 doses etc.) for this population.

Comments on the Quality of English Language

Improved writing, but still need someone to reread this paper who is highly skilled in English. 

 There are typos (no "e" in awareness on line 209),

 use of inappropriate words ("insignificance" doesn't make sense on line 220,

 "perceptions", not "perceptions" on line 208),

 words missing (line 331), etc.   

Title does not make sense - please remove "  variants of concern against".

Title should say multicenter, not multicenters 

Please review the entire manuscript again for errors such as these 

This survey consists of three parts, Table 2 represents the results from the last part which was adapted from Vaccination Attitudes Examination (VAX) Scale https://www.vax-scale.com/services/. In addition, targeted population in this study was healthcare workers who should have the minimum knowledge and awareness to answer these questions.) Scale https://www.vax-scale.com/services/.

This study aimed to understand and evaluate the healthcare workers’ perceptions against COVID-19 booster doses regardless of their vaccination status, therefor vaccine coverage rate was not discussed extensively

Adjusted and modified accordingly

Corrected

Removed

Corrected

Done

Corrected

Done

Thanks for the comment,

Done for the whole manuscript as indicated by the word tracking system

Round 3

Reviewer 1 Report

Thanks for updating the manuscript.